# Potential Urine Proteomic Biomarkers for Focal Segmental Glomerulosclerosis and Minimal Change Disease

**DOI:** 10.3390/ijms232012607

**Published:** 2022-10-20

**Authors:** Natalia V. Chebotareva, Anatoliy Vinogradov, Alexander G. Brzhozovskiy, Daria N. Kashirina, Maria I. Indeykina, Anna E. Bugrova, Marina Lebedeva, Sergey Moiseev, Evgeny N. Nikolaev, Alexey S. Kononikhin

**Affiliations:** 1Nephrology Department, Sechenov First Moscow State Medical University, Trubezkaya, 8, 119048 Moscow, Russia; 2Department of Internal Medicine, Lomonosov Moscow State University, GSP-1, Leninskie Gory, 119991 Moscow, Russia; 3Institute of Biomedical Problems—Russian Federation State Scientific Research Center, Russian Academy of Sciences, Khoroshevskoe Shosse, 76A, 123007 Moscow, Russia; 4Skolkovo Institute of Science and Technology, Bolshoy Boulevard 30, Bld. 1, 121205 Moscow, Russia; 5Emanuel Institute for Biochemical Physics, Russian Academy of Science, Kosygina Str., 4, 119334 Moscow, Russia

**Keywords:** urine proteome, FSGS, MCD, podocyte dysfunction, mass spectrometry

## Abstract

Primary focal segmental glomerulosclerosis (FSGS), along with minimal change disease (MCD), are diseases with primary podocyte damage that are clinically manifested by the nephrotic syndrome. The pathogenesis of these podocytopathies is still unknown, and therefore, the search for biomarkers of these diseases is ongoing. Our aim was to determine of the proteomic profile of urine from patients with FSGS and MCD. Patients with a confirmed diagnosis of FSGS (n = 30) and MCD (n = 9) were recruited for the study. For a comprehensive assessment of the severity of FSGS a special index was introduced, which was calculated as follows: the first score was assigned depending on the level of eGFR, the second score—depending on the proteinuria level, the third score—resistance to steroid therapy. Patients with the sum of these scores of less than 3 were included in group 1, with 3 or more—in group 2. The urinary proteome was analyzed using liquid chromatography/mass spectrometry. The proteome profiles of patients with severe progressive FSGS from group 2, mild FSGS from group 1 and MCD were compared. Results of the label free analysis were validated using targeted LC-MS based on multiple reaction monitoring (MRM) with stable isotope labelled peptide standards (SIS) available for 47 of the 76 proteins identified as differentiating between at least one pair of groups. Quantitative MRM SIS validation measurements for these 47 proteins revealed 22 proteins with significant differences between at least one of the two group pairs and 14 proteins were validated for both comparisons. In addition, all of the 22 proteins validated by MRM SIS analysis showed the same direction of change as at the discovery stage with label-free LC-MS analysis, i.e., up or down regulation in MCD and FSGS1 against FSGS2. Patients from the FSGS group 2 showed a significantly different profile from both FSGS group 1 and MCD. Among the 47 significantly differentiating proteins, the most significant were apolipoprotein A-IV, hemopexin, vitronectin, gelsolin, components of the complement system (C4b, factors B and I), retinol- and vitamin D-binding proteins. Patients with mild form of FSGS and MCD showed lower levels of Cystatin C, gelsolin and complement factor I.

## 1. Introduction

Primary focal segmental glomerulosclerosis (FSGS) and minimal change disease (MCD) are diseases with primary damage to podocytes (primary podocytopathies), clinically manifested by high proteinuria and nephrotic syndrome [1,2]. FSGS is characterized by the presence of sclerosis in parts (segmental) of at least one glomerulus (focal) in a kidney biopsy specimen, when examined by light microscopy, immunofluorescence, or electron microscopy. Minimal change disease (MCD) is the leading cause of the nephrotic syndrome in children (approximately 90 percent) and in a minority of adults (approximately 10 percent) [3]. Light microscopy in case of MCD shows only a minor abnormality in the glomeruli, immunohistological methods display no deposits of immunoglobulins and complements, and electron microscopy reveals a diffuse loss of podocyte foot processes. Development of the nephrotic syndrome is due to the damage of the podocyte, foot process effacement and detachment of the podocyte from the glomerular basement membrane (GBM). As a result, proteins pass through the defects of the GBM and proteinuria develops. The onset of both diseases—FSGS and MCD, is usually acute with a severe nephrotic syndrome. A decrease in kidney function at the onset of the disease is diagnosed in 25–50% of patients, hematuria in 50%, and arterial hypertension in 20% of patients with FSGS [2,4].

Patients with MCD, as well as some patients with FSGS, respond well to steroid therapy [5]. However, 25–50% have steroid resistance—a severe form of FSGS. Severe FSGS is characterized by high proteinuria, renal impairment in the initial stages, and an unfavorable prognosis on the progression of renal dysfunction [2,4,6,7,8].

In primary FSGS, a putative circulating factor that is toxic to the podocyte causes generalized podocyte dysfunction. Secondary FSGS generally occurs as an adaptive phenomenon due to the reduction of the nephron mass or direct toxicity from drugs or viral infections. The circulating factor in FSGS and MCD is still unknown, and therefore the study of specific mechanisms that are involved in podocyte damage is ongoing. This knowledge could improve our understanding of the pathogenetic mechanisms of these diseases. Approaches based on mass spectrometry (MS) are the most objective and sensitive tools that have already provided most of the currently known information on the content of peptides and proteins in urine in various nephropathies [9,10,11,12]. The urinary proteome contains mainly (up to 70%) proteins and peptides of renal origin [13,14]. In general, this approach is the most appropriate for the search for potential biomarkers and mechanisms related to the development and progression of kidney diseases.

The aim of our study was to characterize changes in the urinary proteomic profiles of patients with different course of focal segmental glomerulosclerosis and minimal change disease to determine their specific biomarkers.

## 2. Results

### 2.1. Optimization of the Urine Preparation Protocol for Proteomic Analysis

All clinical urine samples were characterized by proteinuria of varying severity (Table 1). In order to select the optimal method of urine sample preparation for LC-MS/MS analysis 3 previously published methods for concentrating, purifying and hydrolyzing proteins were tested: (1) precipitation of proteins with ice-cold acetone [15]; (2) concentration and hydrolysis of proteins on filters (filter-aided sample preparation (FASP), Microcon (Millipore) filters were used) [16]; (3) ultrafiltration to purify proteins from low molecular weight components of urine [17]. The main criteria for the optimization of urine sample preparation were the robustness and ease of reproducibility of all steps; and the second the effectiveness of the protocol in the view of the number of detected proteins. It was decided not to use the third method due to its excessive laboriousness and poor reproducibility of the ultrafiltration stage for urine samples with proteinuria.

Comparison of the two remaining methods of sample preparation showed that more different proteins was detected using the acetone precipitation method (Table 2, Figure 1). The first protocol allowed to detect the highest number of proteins in the test samples with proteinuria (5, 7, 10 mg/mL of total protein) and was used for further studies with minor modifications.

### 2.2. Label Free Analysis of the Urine Proteome for Patients with FSGS and MCD

Comparison of proteomic profiles of patients with FSGS and and MCD showed no significant differences in the protein levels (Figure 2).

However, the FSGS group in total showed a high variability between the patients inside the group. Thus for a comprehensive assessment of this cohort, a special index was introduced, which was calculated as follows: the first score was assigned depending on the level of eGFR, the second—depending on the severity of proteinuria, the third—steroid resistance of the nephrotic syndrome. Steroid-resistance was defined as the absence of a decrease in proteinuria levels after 16 weeks of prednisolone therapy or a decrease by less than 50% of the baseline level.

The renal function was considered “saved”, if the estimated glomerular filtration rate, determined by the CKD-EPI formula (eGFR CKD-EPI), was above 60 mL/min/1.73 m^2^; and “impaired”—if it was less.

Using this index the patients with FSGS were subdivided into two groups: with a sum of scores of less than 3—mild FSGS (1), and with 3 or higher—severe progressive FSGS (2) (Table 1).

These two subgroups did not differ significantly in the severity of the nephrotic syndrome and renal dysfunction at the onset of the disease. However, in the follow-ups, the patients of the second group were characterized by a more severe FSGS course, meaning impaired renal function and steroid resistance. A wide range of urine proteins was detected at elevated levels in group 2 (Figure 3). For example, an increase in urinary excretion of complement components C3, C4B, factor B, as well as components of the membrane attack complex C8a and C9 were found. The detection of retinol-binding protein 4 and vitamin D-binding proteins in the urine is a consequence of tubulo-interstitial inflammation and injury of the tubular epithelium secondary to glomerular proteinuria [18,19,20,21]. Simultaneously with the interstitial inflammation, the accumulation of extracellular matrix (ECM) components and tubulo-interstitial fibrosis are also activated. Alpha-2-HS-glycoprotein can be attributed to the group of proteins responsible for active processes of ECM accumulation, expression of receptors on cells, and ECM protein metabolism (Figure 3). 

Considering patients with FSGS separately, we found some minor differences in the protein profiles of patients with saved and impaired renal function. In particular, patients with impaired renal function showed higher levels of thyroid hormone-binding protein, β2-microglobulin, vitamin D-binding protein, alpha-2-HS-glycoprotein (fetuin A) (Figure 4).

Proteins that differ between FSGS group 2 and MCD are by 83% identical to those differentiating the two FSGS groups (Figure 5). It can also be seen that like FSGS group 1 patients samples from MCD patients also have elevated levels of osteopontin and the inhibitor of phosphoinositide-3 kinase, while complement proteins, apolipoproteins, hemopexin, vitronectin, and other proteins in urine remain low (Figure 6).

Results of the label free analysis were validated using targeted LC-MS based on multiple reaction monitoring (MRM) with stable isotope labelled peptide standards (SIS) available for 47 of the 76 proteins identified as differentiating between at least one pair of groups (Appendix A). Quantitative MRM SIS validation measurements for these 47 proteins revealed 22 proteins with significant differences between at least one of the two group pairs and 14 proteins were validated for both comparisons (Table 3). Also all of the 22 proteins validated by MRM SIS analysis showed the same direction of change as at the discovery stage with label-free LC-MS analysis, i.e., up or down regulation in MCD and FSGS1 against FSGS2. Moreover, it is worth to note that the absolute values of the measured proteins fold changes between groups for the two quantitation methods (label-free vs. MRM SIS) in their orders of magnitude are in good agreement (Appendix A). 

The most important function and source of proteins are presented in Table 4. The levels of the most significant proteins in arbitrary units in FSGS group 1, FSGS group 2 and MCD are shown in Figure 7, Figure 8 and Figure 9.

## 3. Discussion

In the present study a wide variety of proteins was identified and quantitated using two mass-spectrometric approaches in urine samples of patients with FSGS and MCD. No significant differences were found between the proteomic profiles of patients with MCD and a general FSGS group. However, we found that the differences in the urinary protein profiles in FSGS patients were highly dependent on the severity of the disease—thus the patients were subdivided into those with mild FSGS—normal kidney function and steroid-sensitive NS, and those characterized by severe proteinuria, impaired renal function and steroid-resistant NS. These two groups showed significant differences in the levels of 68 proteins. The group with a severe progressive FSGS showed a number of abundant kidney-derived proteins in urine.

Apolipoproteins, vitronectin, hemopexin and gelsolin reflect the process of active damage to podocytes [24,25,26,32,34,38,39,40,46,47,48,49,50,51]. Apolipoprotein A-1 (ApoA-1), a small 28 kDa high-density lipoprotein (HDL) component, is considered as a putative permeability factor [32,34,50]. In a study by Puig-Gay N. et al. an increase in ApoA-1b in the urine was noted immediately before the onset of proteinuria in patients with FSGS recurrence in the transplant [32]. This protein is absent in the urine of healthy individuals and in most patients with glomerular proteinuria caused by other glomerulopathies [34,50]. Our data also suggests a potential role of ApoA in the pathogenesis of primary FSGS. In terms of response to therapy, our data is consistent with that of Kalantari et al. who differentiated steroid-sensitive and steroid-resistant patients with a confirmed FSGS by urine proteome analysis. Among 21 proteins, ApoA-1 was one of the most significant marker between steroid-sensitive and steroid-resistant forms. An increase in ApoA-1 is associated with hyperlipidemia and low-density lipiprotein oxidation [51].

Like Kalantari et al. and other groups of researchers, we established the role of ApoA-1 and some other urinary proteins in diagnosis of FSGS [32,51,52], however, we took a different approach in the study and introduced a specific index that allows us to evaluate not only the response to steroid therapy, but also and the level of proteinuria and kidney function. This approach has been more effective in separating FSGS patients by urinary proteome to two groups. In addition, the group of patients with mild FSGS is comparable to those with MCD, while the proteomic profile of the FSGS group 2 is significantly different, and apparently, this model can be used to assess the severity of the disease.

Among the biomarkers of podocyte damage, we found an increase in the level of hemopexin, which is a glycoprotein with the highest affinity for hem [24]. Hemopexin is currently considered as one of the possible circulating factors of idiopathic FSGS. Hemopexin binds free methemoglobin, further recognized by CD91 on hepatocytes or macrophages in the spleen, liver, and bone marrow [25]. Cell culture studies have shown that hemopexin can induce the redistribution of the actin cytoskeleton in podocytes and development of proteinuria [24,25,26].

An increase in the urinary excretion of vitronectin in FSGS patients is also of interest, since vitronectin activates β3-integrins, molecules that ensure the fixation of podocytes to the GBM. In case of FSGS, the loss of vitronectin may be associated with the process of podocyte detachment from the glomerular basement membrane [38]. Elevated serum gelsolin aggravates the development of proteinuria and renal dysfunction by F-actin rearrangement, foot process effacement and cell movement [39,40]. These markers of podocyte dysfunction were elevated only in severe steroid-resistant FSGS.

In group 1, patients with mild FSGS, other factors of podocyte dysfunction are detected in the urine—galectin-3—binding protein. It is suggested that this protein is elevated in the kidney tissue in MCD and in the urine in lupus nephritis, and it appears to have a pro-inflammatory function [44,45].

Compared to other glomerulopathies, complement activation processes are not well understood in FSGS. In the second FSGS group we found a significant increase in urinary excretion of complement components C4b, C9, as well as factor B, I and a decrease in CD59, an inhibitor of the membrane attack complex, which indicates the possible role of complement activation associated with stronger damage. Our data is consistent with the results of a study by Huang J. et al., who showed the possibility of systemic complement activation in FSGS patients with increased levels of C3a, C5a, and C5b-9 in blood plasma and urine [43]. Activation of the alternative complement pathway in FSGS may be associated with a poor prognosis [30,31]. On the other hand, an increase in the level of some complement components in the urine may be the result of loss due to severe glomerular filter damage. For a more accurate asessement of complement system disorders in FSGS, the study these components in the blood of these patients is required.

The intensity of accumulation of ECM components is reflected by the excretion of fetuin A, β2 microglobulin, and immunoglobulin chains in the urine of patients with FSGS [37]. The same changes in the protein profiles indicating the damaging processes on tubular cells and accumulation of ECM in the interstitium can be noticed in patients with FSGS with impaired renal function resistant to steroid therapy in our study. An equally important component of disease progression and lack of response to steroid therapy is tubulointerstitial fibrosis and tubular damage in FSGS, which are reflected by an increase in the level in the urine of lumican and cystatin C. These processes are also evidenced by an increase in vitamin D-binding protein and retinol-binding protein 4 in the urine of FSGS patients with impaired renal function [18,19,20,21,37]. Simultaneously with the processes of interstitial inflammation, the mechanisms of accumulation of ECM components are activated [18,19,20]. Our data confirms the results of experimental studies [23,27,28,29,52]. A study of urinary proteins dynamic changes was conducted on a focal segmental glomerulosclerosis rat model (adriamycin-induced nephropathy) and showed that levels of Fetuin-A and alpha 1 microglobulin may be promising markers for early detection of FSGS. Thus, some proteins or their combinations can change with the disease progression [52]. Inter-alpha-trypsin inhibitor heavy chain H1 and H2 activate the CD44+ parietal epithelial cells, the main profibrogenic cells in the glomeruli, and thus are a powerful stimulus for glomerulosclerotic processes [27,28,29].

Many urine proteins are of serum origin and enter the urine through the damaged glomerular filter. However, these proteins can cause additional damage and pathology progression. For example, components of the complement system or plasminogen. The conversion of plasminogen to plasmin in urine can activate the epithelial sodium channels and cause sodium retention in the renal tubules—one of the mechanisms of renal edema [23]. Interestingly, in case of MCD and mild FSGS with a favorable prognosis, some of the proteins found in the urine seem to be of a protective nature, such as for example plasma protease C1 [35].

Perez et al. analyzed urinary peptide profiles using magnetic bead-based technology, combined with MALDI-TOF mass spectrometry, in 44 patients diagnosed with MCD and FSGS. In this work the low molecular weight fraction of urine, containing peptides of proteins were analyzed, while in our work the high-molecular protein fraction was isolated. The authors showed that FSGS patients had higher levels of uromodulin fragments and lower concentrations of fragments of A1AT [53]. In 2017 Perez V. et al. also ran a study that included 24 patients with MCD and 25 patients with FSGS and analyzed their urine proteome using two-dimensional gel electrophoresis in combination with MS to detect urinary biomarkers capable of differentiating MCD and FSGS. They showed that urine concentrations of alpha-1 antitrypsin, transferrin, histatin-3, and 39S ribosomal protein L17 were decreased in the FSGS group, and the calretinin level was increased as compared to the MCD group [54]. We did not find any significant differences between the MCD and the general FSGS groups. However, we found an increase in alpha-1 antitrypsin, and other proteins in patients with severe FSGS as compared with those with MCD and mild FSGS, who showed similar profiles of urine proteins.

Thus, we have identified a wide range of proteins that differ in patients with mild course of steroid-sensitive FSGS/MCD and FSGS patients with a progressive steroid-resistant NS. Proteins excreted in the urine reflect damage to several parts of the nephron—podocytes, tubulo-interstitium, accumulation of ECM, complement activation. Patients with MCD and FSGS with steroid-sensitivity and a favorable course of progress showed similar urine protein profiles, while severe progressive FSGS with steroid-resistant nephrotic syndrome were characterized by the early activation of the complement system and profibrogenic mechanisms and accumulation of extracellular matrix components at the onset of the disease.

## 4. Materials and Methods

### 4.1. Clinical Characteristics of the Patients

Patients with a confirmed diagnosis of FSGS (n = 30) and MCD (n = 9) were recruited for the study, 20 men and 19 women, aged 19 to 69 years, median 37 years [27,57]. The exclusion criteria were: active urinary infection, diabetes mellitus, obesity, severe arterial hypertension, liver disease, rheumatic systemic diseases, stage 5 chronic kidney disease. Impaired renal function (eGFR CKD-EPI < 60 mL/min/1.73 m^2^) was detected in 16 patients, saved kidney function—in 23 (eGFR CKD-EPI > 60 mL/min/1.73 m^2^).

The characteristics of the examined patients are presented in Table 5.

### 4.2. Urine Sample Preparation for LC-MS/MS

First morning urine samples were collected from all examined patients. The middle portion of freshly passed morning urine was collected in 10 mL test tubes, centrifuged at 3000 rpm for 15 min, the supernatant was frozen in 1 mL aliquots and stored at −20 °C.

Urine aliquots with a volume of 0.1 mL were quickly thawed and 0.5 mL of cold acetone was added to precipitate proteins overnight at −20 °C. Then the samples were centrifuged at 20,000× *g* for 10 min, the supernatant was removed, the precipitate was dissolved in 50 µL of 8 M urea/200 mM Tris-HCL, pH 8.5. The proteins were restored with 5 mM dithiotreitol for 30 min at 37 °C, alkylated with 20 mM iodoacetamide in the dark for 30 min. Before hydrolysis, 200 µL of deionized water was diluted, trypsin (Trypsin Gold, Promega, Madison, WI, USA) was added in an enzyme-protein ratio of 1:25, incubated overnight at 37 °C. The reaction was stopped by adding formic acid to the final concentration of 1%. Peptides were centrifuged at 18,000× *g*, the supernatant was left for desalting. Desalting was carried out by solid-phase extraction using plates (Oasis HLB 96-well Microelution Plate, Waters, Beverley, MA, USA). The eluate was lyophilized and dissolved in 0.1% formic acid to a concentration of 0.5mg/mL for further LC-MS/MS analysis.

### 4.3. Label-Free Untargeted LC-MS/MS Urine Proteomic Analysis

The resulting tryptic peptide mixture was analyzed using liquid chromatography coupled with tandem mass spectrometry (LC-MS/MS) method based on a nano-HPLC Dionex Ultimate3000 system (Thermo Fisher Scientific, Madison, WI, USA) and a timsTOF Pro (Bruker Daltonics, Billerica, MA, USA) mass spectrometer. A packed emitter column (C18, 25 cm × 75 μm 1.6 μm) (Ion Optics, Parkville, Australia) was used to separate peptides at a flow rate of 400 nL/min by gradient elution from 4% to 90% of phase B during 40 min. Mobile phase A consisted of 0.1% formic acid in water and mobile phase B consisted of 0.1% formic acid in acetonitrile.

Mass spectrometric analysis was performed using the parallel accumulation serial fragmentation (PASEF) acquisition method. An electrospray ionization (ESI) source was operated at 1500 V capillary voltage, 500 V end plate offset and 3.0 L/min of dry gas at temperature of 180 °C. The measurements were carried out in the m/z range from 100 to 1700 Th. The ion mobility was in the range from 0.60 to 1.60 V s/cm^2^. The total cycle time was 1.88 s and the number of PASEF MS/MS scans was set to 10.


*Targeted quantitative LC-MS/MS using multiple reaction monitoring (MRM) with stable isotope labelled peptide standards (SIS).*


Targeted quantitative LC-MS analysis was carried out using synthetic stable-isotope labeled internal standard (SIS) and natural (NAT) synthetic proteotypic peptides for measurements of the corresponding proteins in urine. The selected 22 SIS and NAT synthetic peptides had been previously validated for use in LC/MRM-MS experiments [55]. LC-MS parameters, such as the LC gradient and the MRM parameters (Q1 and MRM scans) were adapted and optimized based on the previous studies [56]. The SIS peptide mixture was spiked in each urine sample at a balanced concentration which was optimized in experiments with dilution series of urine samples with proteinuria. Standard curves were generated using NAT and SIS peptide standards with a pooled urine sample as matrix.

All samples were analyzed in duplicate by HPLC-MS system consisting of an ExionLC™ (UHPLC system (ThermoFisher Scientific, Waltham, MA, USA) coupled online to a SCIEX QTRAP 6500+ triple quadrupole mass spectrometer (SCIEX, Toronto, ON, Canada). The loaded sample volume was 10 μL per injection. HPLC separation was carried out using Zorbax Eclipse Plus C18 RRHD column (150 × 2.1 mm, 1.8 μm) (Agilent, Santa Clara, CA, USA) with gradient elution. Mobile phase A was 0.1% FA in water; mobile phase B was 0.1% FA in acetonitrile. LC separation was performed at a flow rate of 0.4 mL/min using a 53 min gradient from 2 to 45% of mobile phase B. Mass-spectrometric measurements were carried out using the MRM acquisition method. The electrospray ionization (ESI) source settings were as follows: ion spray voltage 4000 V, temperature 450 °C, ion source gas 40 L/min. The corresponding transition list for MRM experiments with retention times values and Q1/Q3 masses for each peptide were adapted from the previous studies [56].

Skyline Quantitative Analysis software (version 20.2.0.343, University of Washington) was used for quantitative analysis.

### 4.4. Data Analysis

The data obtained were analyzed using PEAKS XPro software (BSI, North Waterloo, ON, Canada) according to the following parameters: parent mass error tolerance −20 ppm; fragment mass error tolerance −0.03 Da; enzyme—trypsin; missed cleavages—3; fixed modifications—Carbamidomethyl (C); variable modifications—Oxidation (M), Acetylation (N-term). The search was carried out using the SwissProt Human database. False discovery rate threshold was set to 0.01. Scripts written in R version 3.3.3 [58] and RStudio 1.383 [57] were used for statistical processing of the results.

## 5. Conclusions

Thus, the proteomic profile of urine from patients with FSGS and MCD is characterized by a large number of excreted proteins, but no significant differences between these two forms were revealed. However, the FSGS patients showed high variability inside the group and clustered into two subgroups, which could be reliably distinguished basing on the proteomic profile. Results of the label free analysis were validated using targeted LC-MS based on multiple reaction monitoring (MRM) with stable isotope labelled peptide standards (SIS) available for 47 of the 76 proteins identified as differentiating between at least one pair of groups. Quantitative MRM SIS validation measurements for these 47 proteins revealed 22 proteins with significant differences between at least one of the two group pairs and 14 proteins were validated for both comparisons. In addition, all of the 22 proteins validated by MRM SIS analysis showed the same direction of change as at the discovery stage with label-free LC-MS analysis, i.e., up or down regulation in MCD and FSGS1 against FSGS2. In patients with severe FSGS 2, the urine proteome panel reflects damage to podocytes (Vitronectin, Hemopexin, Gelsolin, Apolipiprotein A), complement activation (Complement component C4b, C9, Complement factor B and I), and accumulation of the extracellular matrix and tubular damage (Cystatin C, Vitamin D-binding protein, Retinol-binding protein 4, Alpha-2-HS-glycoprotein, Plasma protease C1 inhibitor, Lumican, Clusterin).

## Figures and Tables

**Figure 1 ijms-23-12607-f001:**
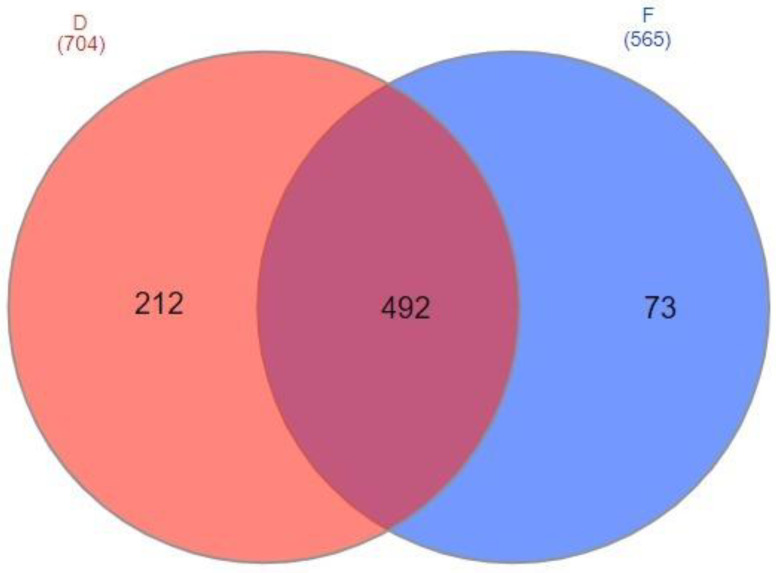
Venn diagram comparing different methods of urine sample preparation for proteomic analysis (D—ice-cold acetone protein precipitation method; F—protein concentration and hydrolysis on filters (FASP)).

**Figure 2 ijms-23-12607-f002:**
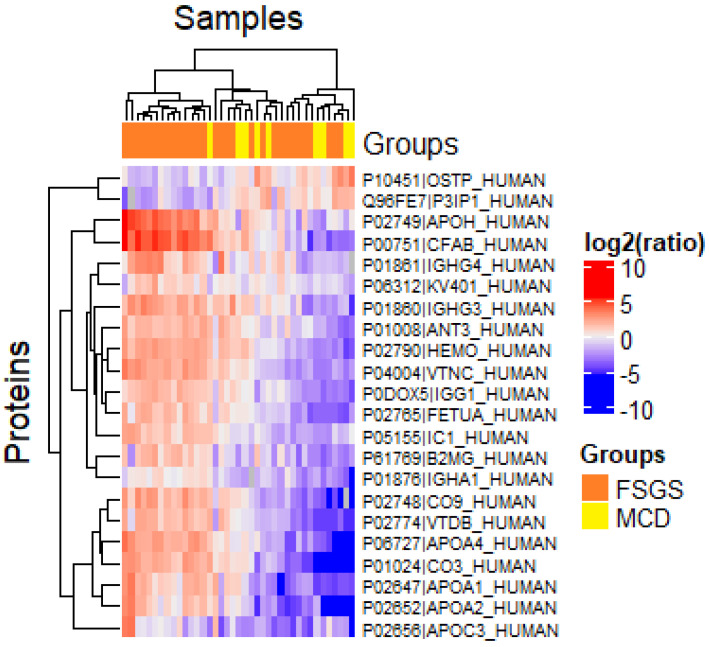
Hierarchical clustering of proteins identified in urine samples of patients with MCD and FSGS. The box denotes log2-transformed values of peak intensity.

**Figure 3 ijms-23-12607-f003:**
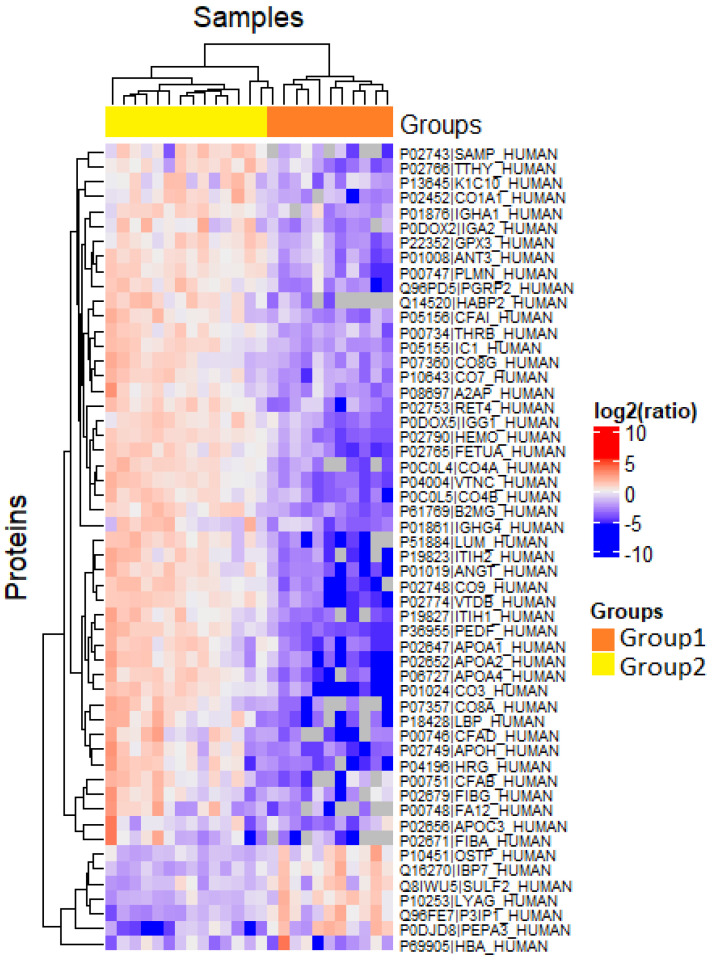
Hierarchical clustering of proteins identified in urine samples of patients with FSGS in group 1 and group 2. The box denotes log2-transformed values of peak intensity.

**Figure 4 ijms-23-12607-f004:**
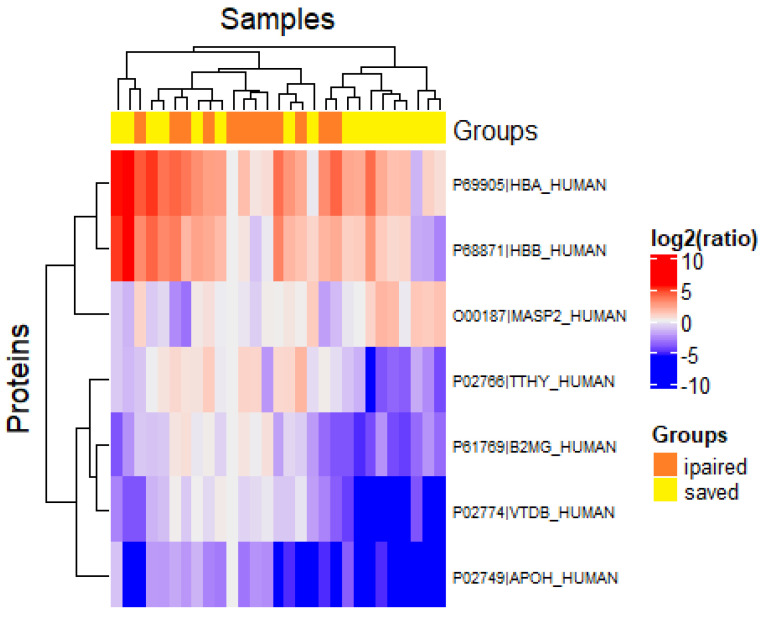
Hierarchical clustering of proteins identified in urine samples of FSGS patients with saved and impaired renal function. The color gradient denotes the log2-transformed ratio of the mean peak intensity values measured in the two groups.

**Figure 5 ijms-23-12607-f005:**
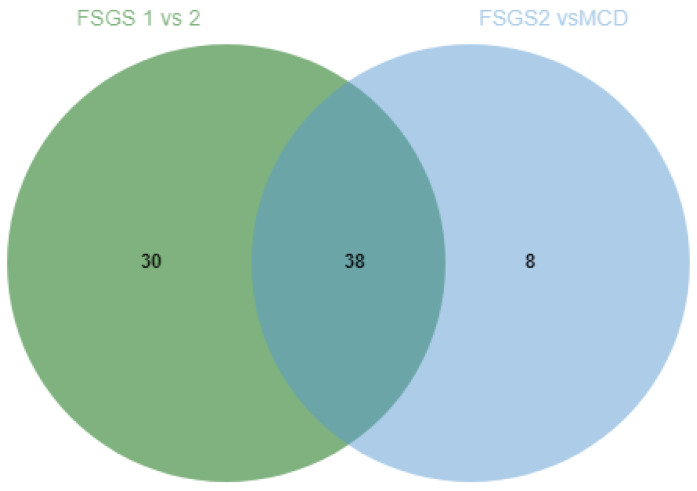
Proteins that differ between FSGS sub groups (1 and 2) and MCD.

**Figure 6 ijms-23-12607-f006:**
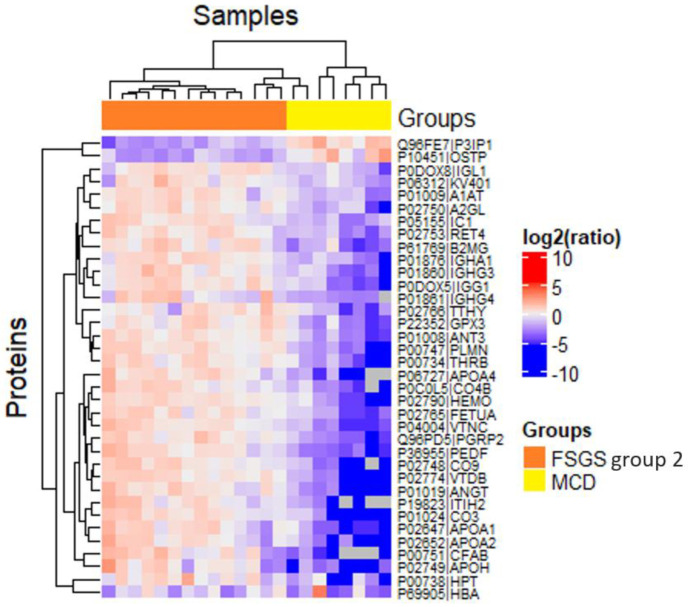
Hierarchical clustering of proteins identified in urine samples of patients with FSGS in group 2 and MCD. The color gradient denotes the log2-transformed ratio of the mean peak intensity values measured in the two groups.

**Figure 7 ijms-23-12607-f007:**
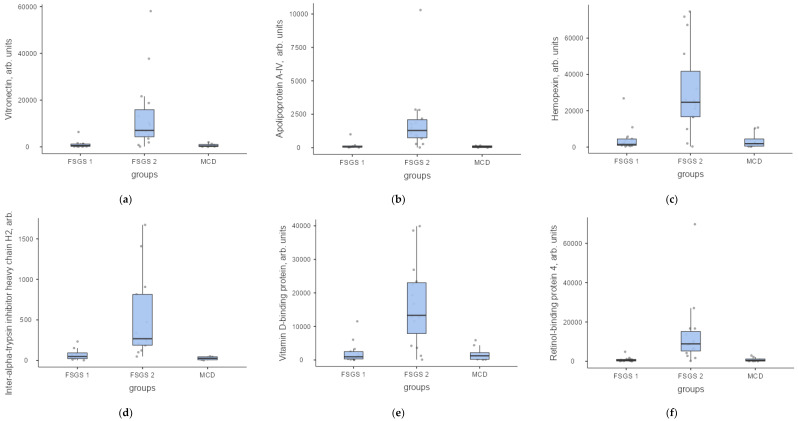
Protein levels in the urine of patients with FSGS (group 1—FSGS 1 and group 2—FSGS 2) and MCD in arb. units: (**a**) Vitronectin, (**b**) Apolipiprotein A-IV, (**c**) Hemopexin, (**d**) Inter-alfa-trypsin inhibitor heavy chain H2; (**e**) Vitamin-D-binding protein; (**f**) Retinol-binding protein.

**Figure 8 ijms-23-12607-f008:**
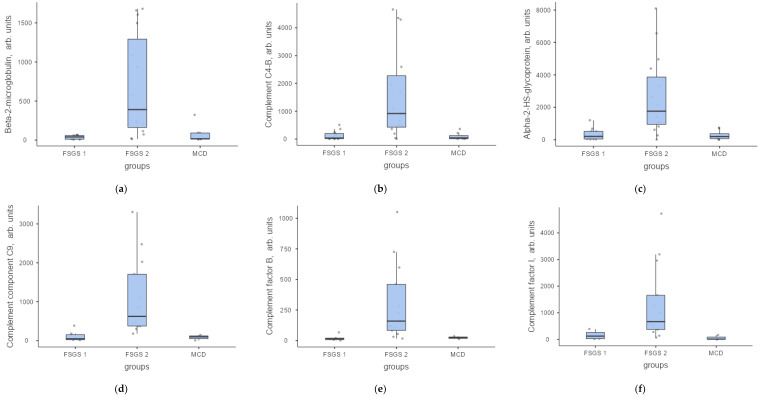
Protein levels in the urine of patients with FSGS (group 1—FSGS 1 and group 2—FSGS 2) and MCD in arb. units: (**a**) beta-2 microglobulin, (**b**) Complement C4-b, (**c**) alfa-2-HS-glycoprotein, (**d**) Complement component C9; (**e**) Complement factor B; (**f**) Complement factor I.

**Figure 9 ijms-23-12607-f009:**
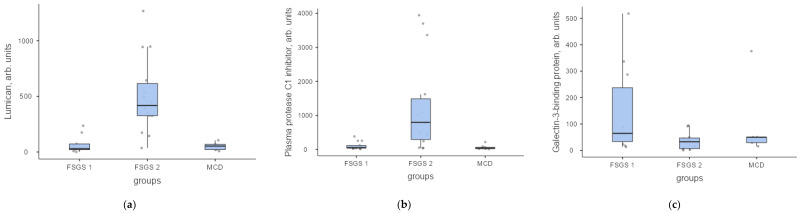
Protein levels in the urine of patients with FSGS (group 1—FSGS 1 and group 2—FSGS 2) and MCD in arb. units: (**a**) lumican, (**b**) plasma protease C1 inhibitor, (**c**) galectin-3-binding protein.

**Table 1 ijms-23-12607-t001:** Calculation of the FSGS severity index.

Parameters	Score
eGFR CKD-EPI, mL/min/1.73 m^2^	
>60	0
45–59	1
35–45	2
<35	3
Proteinuria, g/24 h	
>2	0
2–3	0.5
3–4	1
4–5	1.5
5–6	2
6–7	2.5
>7	3
Steroid resistance	
Absent	0
Present	1

**Table 2 ijms-23-12607-t002:** Efficiency of urine proteins extraction by two methods (Acetone precipitation by ice-cold acetone protein precipitation; FASP—protein concentration and hydrolysis on filters). The number of identified proteins is indicated in Table.

Method	Sample 1 (5 mg/mL)	Sample 1 (7 mg/mL)	Sample 1 (10 mg/mL)
Acetone precipitation	439	520	530
FASP	362	456	441

**Table 3 ijms-23-12607-t003:** Urinary proteins selected in this study for validation as perpective potental biomarkers for differentiating patients with MCD and FSGS (group 1—FSGS 1 and group 2—FSGS 2).

Protein ID		Description	FSGS 1 vs. FSGS 2		FSGS 2 vs. MCD		Direction Change in FSGS 2	Average Fold Change between Groups	Validated in at Least 1 Group	Validated in Both Groups
			TIMS LFQ (Discovery Phase)	QQQ SIS MRM (Validation Phase)	TIMS LFQ (Discovery Phase)	QQQ SIS MRM (Validation Phase)				
			Significant FSGS 1 vs. FSGS 2 (−10 × LOG(p) > 20)	Significant FSGS 1 vs. FSGS 2 (*p* < 0.05)	Significant FSGS 2 vs. MCD (−10 × LOG(*p*) > 20)	Significant FSGS 2 vs. MCD (*p* < 0.05)				
1	P04004	Vitronectin	+	+	+	+	up	10	+	+
2	P06727	Apolipoprotein A-IV	+	+	+	+	up	10	+	+
3	P19823	Inter-alpha-trypsin inhibitor heavy chain H2	+	+	+	+	up	9	+	+
4	P02774	Vitamin D-binding protein	+	+	+	+	up	8	+	+
5	P61769	Beta-2-microglobulin	+	+	+	+	up	8	+	+
6	P0C0L5	Complement C4-B	+	+	+	+	up	7	+	+
7	P02765	Alpha-2-HS-glycoprotein	+	+	+	+	up	6	+	+
8	P02790	Hemopexin	+	+	+	+	up	6	+	+
9	P05155	Plasma protease C1 inhibitor	+	+	+	+	up	6	+	+
10	P02753	Retinol-binding protein 4	+	+	+	+	up	6	+	+
11	P00747	Plasminogen	+	+	+	+	up	6	+	+
12	P00734	Prothrombin	+	+	+	+	up	6	+	+
13	P02766	Transthyretin	+	+	+	+	up	5	+	+
14	P06312	Immunoglobulin kappa variable 4-1	+	+	+	+	up	3	+	+
15	P10909	Clusterin		+	+	+	up	3	+	
16	P02748	Complement component C9	+	+	+		up	11	+	
17	P00751	Complement factor B	+	+	+		up	11	+	
18	P51884	Lumican	+	+		+	up	9	+	
19	P05156	Complement factor I	+	+		+	up	6	+	
20	P01034	Cystatin-C	+	+		+	up	3	+	
21	P06396	Gelsolin	+	+			up	2	+	
22	Q08380	Galectin-3-binding protein	+	+			down	6	+	

**Table 4 ijms-23-12607-t004:** Description of urinary proteins selected in this study for differentiating patients with MCD and FSGS (group 1—FSGS 1 and group 2—FSGS 2).

Protein Group	Description	Clinical/Histological Form	Pathogenetic Role
1	Lumican	FSGS 2 group	Lumican, an extracellular matrix proteoglycan, related to ECM accumulation [22]
2	Vitamin D-binding protein	FSGS 2 group	Potential marker of renal interstitial inflammation and fibrosis, and steroid-resistant nephrotic syndrome [18,19,20]
3	Plasminogen	FSGS 2 group	Plasma abundant protein. Converts to plasmin, may activate epithelial sodium channels causing sodium retention and edema [23]
4	Hemopexin	FSGS 2 group	Hemopexin induces nephrin-dependent reorganization of the actin cytoskeleton in podocytes [24,25,26]
5	Prothrombin	FSGS 2 group	Plasma abundant protein
6	Complement factor I	FSGS 2 group	Plasma abundant protein
7	Inter-alpha-trypsin inhibitor heavy chain H2	FSGS 2 group	The inter-alpha-trypsin inhibitors (ITI) are a family of structurally related plasma serine protease inhibitors involved in extracellular matrix stabilization. ITIs are involved in the accumulation of tubulo-interstitial fibrosis in severe forms of FSGS and it activates CD44 + parietal profibrogenic cells in FSGS [27,28,29]
8	Transthyretin	FSGS 2 group	Plasma abundant protein
9	Complement factor B	FSGS 2 group	The complement components C4B showed a massive increase in protein abundance in FSGS [30,31]
10	Apolipoprotein A-I	FSGS 2 group	ApoA-1b is noted to be present in the urine of recurrent FSGS possibly correlating with disease activity [32,33,34]
	Apolipoprotein A-IV	FSGS 2	Plasma abundant protein
11	Complement component C9	FSGS 2	The complement components C1 and C4B, properdin (CFP) showed a massive increase in protein abundance in FSGS [30]
12	Plasma protease C1 inhibitor	FSGS 2	Activation of the C1 complex is under control of the C1-inhibitor. It forms a proteolytically inactive complex with the C1r or C1s proteases. May play a potentially crucial role in regulating important physiological pathways including complement activation, blood coagulation, fibrinolysis and the generation of kinins. Acute phase marker [35]
14	Alpha-2-HS-glycoprotein/Fetuin A	FSGS 2	In proteinuric patients, significant urinary losses of fetuin-A may cause low serum fetuin-A levels. However, its peptides are elevated in the urine of patients with a high percentage of TIF [36,37]
15	Retinol-binding protein 4	FSGS 2	It is filtered through the GBM and reabsorbed in the tubules, reflecting lysosomal proteolysis in the tubular epithelium. Its increase primarily indicates tubular damage. In addition, its level is associated with response to therapy. [21]
16	Vitronectin	FSGS 2	Vitronectin activates integrins, through which podocytes are attached to the GBM. Possibly vitronectin activation is involved in podocyte detachment from GBM [38]
17	Beta-2-microglobulin	FSGS 2	Plasma abundant protein
18	Immunoglobulin kappa variable 4-1	FSGS 2	Plasma abundant protein
19	Gelsolin	FSGS 2	Gelsolin, a Ca-dependent actin-binding protein, induces a change in the orientation of the actin filament, indicating a conformational change in actin [39,40]
20	Cystatin	FSGS 2	Urinary cystatin C as a specific marker of tubular dysfunction [41]
21	Clusterin	FSGS 2	Clusterin facilitates in vivo clearance of extracellular misfolded proteins and apoptosis. Clusterin has been postulated as a down modulator of the inflammatory response [42]
22	Complement C4-B	FSGS 2	The complement components C1 and C4B, properdin (CFP) showed a massive increase in protein abundance in FSGS [30,43]
23	Galectin-3-binding protein	FSGS 1 and MCD	Galectin-3-binding protein is a secreted, hyperglycosylated protein expressed by the majority of human cells. Urinary G3BP is a non-invasive biomarker for clinically and histologically reflecting lupus nephritis activity [44,45]
total 22 proteins			

**Table 5 ijms-23-12607-t005:** Characteristics of the patients.

Parameters	FSGS ^1^ (n = 30)	MCD ^2^ (n = 9)
Age, years	40 (27.3; 57.8)	35 (28; 59)
Gender (male), n (%)	18 (60)	2 (22.2)
Arterial hypetension, n (%)	22 (73.3)	2 (22.2)
Proteinuria, g/24h	3.66 (2.50; 5.00)	3.24 (2.03; 3.5)
Serum albumin, g/L	26.55 (20.85; 33.68)	29.3 (20.00; 35.80)
Serum protein, g/L	50.8 (40.86; 58.23)	61.4 (46.5; 65.3)
Nephrotic syndrome, n (%)	21 (70)	9 (100)
Creatinine, mkmol/L	109.31 (77.57; 152.65)	85.9 (71.8; 115.9)
eGFR^2^ CKD-EPI ^3^,mL/min/1.73 m^2^	64.68 (41.4; 97.09)	73 (55.58; 105.00)
eGFR< 60 mL/min/1.73 m^2^, n (%)	12 (40.0)	4 (44.4)
Steroid-resistant NS, n (%)	14 (46.7)	0

^1^ Focal segmental glomerulosclerosis, ^2^ Minimal change disease, ^3^ Estimated glomerular filtration rate using the CKD-EPI formula. The table shows the median, in brackets—the 1st and 3rd quartiles.

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
