# Peer review of "Potential Urine Proteomic Biomarkers for Focal Segmental Glomerulosclerosis and Minimal Change Disease"

_ijms, 2022, doi:10.3390/ijms232012607_

Round 1

Reviewer 1 Report

The authors seek for FSGS and MCD biomarkers in patients’ urine using proteomics approach.

Comments

1.      The language of the manuscript should be improved. All the typos should be corrected. All abbreviations should be disclosed at first use.

2.      Lines 23-26: The authors should indicate here, which subgroup of FSGS patients has more severe disease.

3.      Introduction: The authors should describe in more detail each condition (mild FSGS, severe FSGS, MCD).

4.      Lines 64-66: The aim of the study should be rephrased as it does not correspond to the Results obtained.

5.      Lines 69-82: The authors should indicate the criteria for the optimization of urine sample preparations.

6.      Line 89: The authors should designate the forms of FSGS.

7.      Lines 95-99: The authors should unify group names: mild- severe- progressive FSGS or primary-secondary FSGS, or group1-group 2 FSGS etc.

8.      Lines 105-109: these dada is related to Discussion section and should be moved there. Reference or explanation are required for this statement.

9.      Line 115, 152, 154, 208: Reference is required at the end of the sentence.

10.  Table 3 is missing. This should be corrected.

11.  Line 142-143: “ a large number of proteins” compared with what? This should be clarified.

12.  Line 146: The response to therapy was not explored in the study. This should be corrected.

13.  Line 163: Previous study [REF # 17] profoundly investigated the same subject and obtained similar results in part. The authors should make a comparison of the both studies, explain disparity obtained in their study if any, and prove the novelty of their research.

14.  Lines 198-200: The authors should present rationale for their suggestion.

15.  Lines 254-258: As remission was not explored in the study this paragraph should be removed.

16.  Section 4.2 and 4.3 are identical. This should be corrected.

17.  Conclusions: The authors should indicate which of the examined proteins could be used as urine proteomic biomarkers while their prognostic value should be estimated using ROC analyses.

Author Response

Dear Reviewer,

Thank you for your review, important comments and recommendations.

Best regards,

Natalia Chebotareva and co-authors

Reviewer 2 Report

The submitted study describes the proteomic analysis of urine samples collected from focal segmental glomerulosclerosis (FSGS) and minimal change disease (MCD) patients to discover biomarkers that characterize these diseases. The finding of the submitted study are considered informative and therefore it would be appropriate for publication in International Journal of Molecular Sciences. However, there are some points to be considered. Please consider and revise the manuscript according to the comments listed below.

(1)   In 2014, Pérez, et al. reported a urinary peptide profiling to compare MCD and FSGS. They mentioned that the peak areas corresponding to uromodulin and α-1-antitrypsin fragment showed higher and lower expressions, respectively in FSGS patients compared to MCD patients (PLoS One. 2014;9(1):e87731). In another study reported in 2017 (Pérez, et al., BMC Nephrol. 2017;18(1):49), they concluded that urinary concentration of α-1 antitrypsin, transferrin, histatin-3 and 39S ribosomal protein L17 was decreased and calretinin was increased in FSGS compared to MCD. Please cite these papers and then add a discussion of the differences in methodology and results between Pérez, et al.'s study and the submitted study.

(2)   Criteria for classification of steroid resistance (3rd score) are not clear; please add a detailed explanation of how the patient is determined to be steroid-resistant NS from the description in the section 4.1.

(3)   Similar to the above comments, the criteria for classifying the patients with saved and impaired renal function (Figure 4) are not clear.

(4)   Urinary protein levels should fluctuate throughout the day. Was the urine collection always at the same time of day?

(5)   The following are some other minor points.

・The caption of Figure 3 shows "Group 1" and "Group 2," while the figure shows "0" and "1.

・The caption of Figure 4 shows "impaired", while the figure shows "NotSaved.

・Most of the content of the sections 4.2 and 4.3 was duplicated.

・Table 2 on page 18 would be probably Table 4.

Author Response

Dear Reviewer,

Thank you for your important suggestions to improve the manuscript.

Best regards,

Natalia Chebotareva and co-authors

Reviewer 3 Report

The article entitled ' Potential Urine Proteomic Biomarkers for Focal Segmental Glomerulosclerosis and Minimal Change Disease’ greatly describes the potential biomarkers for Focal Segmental Glomerulosclerosis and Minimal Change Disease. The article is very interesting and needs further evidence.  However, The authors could improve the article.

1. The data provided is very preliminary and the article needs further exploration and confirmation of the Biomarkers proposed in the study. Some molecular assays are required to confirm them.

2. The authors also mentioned that there are no significant differences in the urine proteomic profile, the authors need to consolidate their findings with additional orthogonal methods.

3. The study mainly provides Bioinformatic analysis data for the disease. The Biomarkers are of potential interest and need to be confirmed by other methods.

4. Review of English language is also required for the article.

Author Response

Dear Reviewer,

Thank you for you comments. We tried to answer all  questions.

Best regards,

Authors

Reviewer 4 Report

In the present study, the authors compared the urine proteomes from patients with severe or mild primary focal segmental glomerulosclerosis and with minimal change disease. Although the results contain useful information, the manuscript as a whole is descriptive and lacks inspirable messages for the readers. The reviewer cannot recommend this manuscript as a candidate for publication in IJMS journal in the current version.

General issues:

The Conclusions section is a set of general phrases and raises more questions than answers.

References sometimes do not meet the criteria established in the journal's “Instructions for Authors”.

And finally, authors need to do a spell check of their manuscript as there are some mistakes throughout the text. The absence of an abbreviations list also complicates the perception of the material.

line 204 "a high-density lipoprotein (LDL)" - why LDL?

line 285 is partially in Russian.

Author Response

Dear Reviewer,

Thank you for comments!

Best regards,

Authors

Round 2

Reviewer 4 Report

The manuscript was improved significantly according the reviewer's comments so that readers can obtain useful message from the research by authors.